# Genome-Wide Identification and Expression Analysis of MYC Transcription Factor Family Genes in Rubber Tree (*Hevea brasiliensis* Muell. Arg.)

**Shi-Xin Zhang, Shao-Hua Wu, Jin-Quan Chao, Shu-Guang Yang, Jie Bao and Wei-Min Tian ***

Key Laboratory of Biology and Genetic Resources of Rubber Tree, Ministry of Agriculture and Rural Affairs Ministry of Agriculture, State Key Laboratory Incubation Base for Cultivation and Physiology of Tropical Crops, Rubber Research Institute, CATAS, Haikou 571100, China; sxzhang@catas.cn (S.-X.Z.); wush-rri@catas.cn (S.-H.W.); chaojinquan@catas.cn (J.-Q.C.); yangshuguang@catas.cn (S.-G.Y.); allen-bao-1985@outlook.com (J.B.)
* Correspondence: wmtian@catas.cn; Tel.: +86-898-2330-0309; Fax: +86-898-2330-0315

**Abstract:** Myelocytomatosis (MYC) transcription factors play a core regulator in the jasmonic acid signaling pathway, which regulates the secondary laticifer differentiation and rubber biosynthesis in rubber tree (*Hevea brasiliensis*). However, there are currently no reports on the MYC gene family in rubber trees, an important industrial raw material crop worldwide. In the present study, 32 *HblMYCs* were isolated and identified. The diversity in gene structure and presence of various cis-regulatory elements in promotors suggest that *HblMYCs* participate in various biological processes. Based on the expression patterns in the cambium region and laticifer in, respectively, response to coronatine (COR) and tapping, and the phylogenetic relationship with the MYCs that have been functionally identified in other plants, the *HblMYC24* and *HblMYC30* may be related to laticifer differentiation while the *HblMYC6*, *HblMYC11* and *HblMYC15*, as well as *HblMYC16* and *HblMYC21*, may positively regulate rubber biosynthesis. The results provide a foundation for understanding the molecular mechanism of jasmonate signaling in regulating laticifer differentiation and rubber biosynthesis in rubber tree.

**Keywords:** *Hevea brasiliensis*; MYC transcriptional factor; jasmonate; laticifer differentiation; rubber biosynthesis



## 1. Introduction

The myelocytomatosis (MYC) transcription factors are characterized by an HLH-MYC_N domain in the N-terminal, a HLH (basic helix-loop-helix) domain in the C-terminal and at least one nuclear localization signal (NLS) [1]. The MYCs are the crucial components of the COI1/JAZ/MYC complex in jasmonate (JA), signaling by transcriptionally activating the JA-responsive genes. The binding of JA-Ile to the JAZ-COI1 complex causes the JAZ protein to be excised and degraded by the 26S proteasome, and the release of the MYC2 [2,3]. Thereafter, MED25 links COI1 with HAC1-dependent H3K9 acetylation to activate MYC2's transcriptional regulation of JA-responsive genes [3,4]. Recently, the LEUNIG_HOMOLOG(LUH), a component of a new MYC2-MED25 functional complex, has been found to act as a scaffold protein to participate in MYC2 by integrating MED25 and HAC1 [5].

The rubber tree (*Hevea brasiliensis* Muell. Arg.) is the primary source of natural rubber (NR) in the world. The NR is synthesized and stored in laticifers. The number of the secondary laticifers in the trunk bark is closely related to the rubber yield. The secondary laticifer is differentiated from vascular cambia (Hao and Wu, 2000 [6]), and induced by such factors as jasmonic acid (JA) [6–8], mechanical wounding [7], and coronatine (COR) [9–11], an active jasmonate homolog [12] Available data show that both the secondary laticifer differentiation [6,13] and the rubber biosynthesis [14] are positively controlled by JA signalling. Therefore, the MYC members in rubber trees should be involved in regulating

such processes. Indeed, the *HbMYC2* [14] and *HbMYC2b* [15] have, respectively, identified to transcriptionally activate the expression of the farnesyl pyrophosphate synthase gene (*HbFPS*) and a small rubber particle protein gene (*HbSRPP*), two important proteins for NR biosynthesis. Nevertheless, the identified MYC homologues are still limited [16]. It is necessary to identify MYC transcription factors genome-wide in rubber trees with the aid of a reference genome [17]. In the present study, we performed a genome-wide identification of MYC members in rubber trees, analyzed their expression patterns and screened the candidate MYC members that were related to the secondary laticifer differentiation and ruber biosynthesis.

## 2. Methods

### 2.1. Plant Materials

Three materials of the rubber tree clone CATAS7-33-97, etc. epicormic shoots from the grafting buds, eight-year-old virgin trees, and tapping trees, were grown on the experimental base of Rubber Research Institute, Experimental Farm of the Chinese Academy of Tropical Agricultural Sciences (CATAS) on Hainan Island, PR China. The epicormic shoots were grown from the latent buds and flushed five to six times in the current year. The tapping trees which had been regularly tapped for two years.

### 2.2. Analyses of Tissue-Specific Expression

The feeder root (Root); stem at bronze stage (Stem); mature leaf (Leaf); female flower (Flower); perisperm of seed (Perisperm); trunk bark (Bark); cambium region of stem at stationary phase (Cambium); and latex of the virgin tree (Latex) of CATAS7-33-97 were collected without any treatments.

### 2.3. Coronatine (COR) Treatment and Cambium Tissue Isolation

When the epicormic shoots had developed more than two extension units (EU), treatments were performed on EU1 [13]. The stem surface of 2 cm × 2 cm in the middle of the EU1 was scraped with a sharp razor to remove the epidermis cuticle and the part of the cortex. 240 shoots were divided into two groups. One group included 120 shoots and their wounded surfaces were wrapped with a polyethylene membrane soon after being applied with 20 μM COR (Sigma Aldrich (Shanghai) Trading Co., LTD, Article number: C8115, China) [10,11]. As control, the other was directly wrapped with a polyethylene membrane. The bark samples, including phloem, cambium, and xylem, were collected by scraping with an RNase-free sharp razor at 0.5 h; 1 h; 2 h; 4 h; 8 h; 1 d; 2 d; and 3 d after COR treatment. Bark samples were collected from 15 shoots at each time interval. The cambium tissues that were isolated by the tangential section of frozen section technique from the 15 shoots were then mixed into the pyrolysis buffer SL (RNA prep Pure Plant Kit, TIANGEN BIOTECH (BEIJING) CO., LTD., Beijing, China) on ice and stored at −80 °C until RNA extraction.

### 2.4. Tapping Treatment and Latex Collection

A total of 140 eight-year-old virgin trees were used as experimental materials and divided into two groups. Each group included 70 trees. One group was treated by tapping while as control, the other without any treatment. The latex samples were, respectively, collected at 0.5 h; 2 h; 6 h; 12 h; 1 d; 2 d; and 3 d after tapping treatment from ten trees at each time interval. As control, the latex samples were collected from the trees without any treatment at corresponding time intervals. Latex samples were directly mixed into the pyrolysis buffer SL (RNA prep Pure Plant Kit, TIANGEN BIOTECH (BEIJING) CO., LTD., Beijing, China) on the ice and stored at −80 °C until RNA extraction.

### 2.5. Total RNA Isolation and cDNA Synthesis

Total RNA was extracted from rubber tree tissues and latex according to the method of RNAprep Pure Plant Kit (TIANGEN BIOTECH (BEIJING) CO., LTD., Beijing, China) and degraded with RNase-Free DNase I (TIANGEN BIOTECH (BEIJING) CO., LTD., Beijing,

China), according to the manufacturer's protocol. The RNA quantity and quality were determined by a spectrophotometer (Thermo Fisher Scientific, Inc., Waltham, MA, USA).

A total of RNA (1 μg) was reversely transcribed in a PCR amplifier and the first strand of cDNA was synthesized with the Thermo Scientific Revert Aid First Strand cDNA Synthesis (Thermo Fisher Scientific Inc., MA, USA).

### 2.6. Amplification of the Open Reading Frame (ORF)

Firstly, we obtained 200 scaffolds with the bHLH domain based on the MYC homologues from the latex transcriptome data of rubber tree clone CATAS 8-79 and PR107 [18], the cambium transcriptome data (Wu 2016), and the genome of rubber tree clone CATAS 7-33-97 [17]. Secondly, we obtained 32 MYC members by the specific MYC domains [1] and removing the incomplete sequences. Thirdly, we cloned the full length cDNAs of MYC members with specific primers by RACE according to SMARTer™ RACE cDNA Amplification Kit User Manual (Clontech Laboratories, Inc., CA, USA). Amplification reactions of the 3′- and 5′-RACE were performed with a thermal cycling profile of 94 °C for 3 min and 30 cycles at 94 °C for 30 s; 65 °C for 30 s; and 72 °C for 2.5 min, followed by an extension at 72 °C for 20 min in the PCR Amplifier. The PCR products were cloned into pMD-18 T vector (TAKARA BIO INC., Dalian, China) and sequenced (Life Technologies, Guangzhou, China). The open reading frames (ORF) were testified by RT-PCR with special primers listed in Table S1. The amino acid sequences of the cloned cDNA fragment were deduced, and protein sequences were aligned using the software of DNAMAN 8.0.

### 2.7. Conserved Domain Analysis and Phylogenetic Tree

The bHLH-MYC_N domain (Accession: pfam14215) and HLH domain (Accession: cd00083) were analyzed by using the programs of NCBI Conserved Domains and SMART (http://smart.embl-heidelberg.de/smart/set_mode.cgi?NORMAL=1, accessed on 14 February 2022). The Nuclear localization signal (NLS) was detected by using the SeqNLS (http://mleg.cse.sc.edu/seqNLS/, accessed on 14 February 2022).

To analyze the evolutionary relationships and infer their putative function, a neighbor-joining tree between 32 *HblMYCs*; 13AtMYCs; 27 TaMYCs [19]; 23 SsMYCs [20]; and 59 other putative MYCs of a total of 22 plants in Phytozome v13 (https://phytozome-next.jgi.doe.gov/, accessed on 14 February 2022) were constructed by the software of MEGA7.0 (Amino acid sequence of 32 *HblMYC* proteins and other 122 MYC proteins in Table S2). The robustness of the phylogram's topology was determined by a bootstrap analysis (1000 replicates).

### 2.8. Real-Time PCR Analysis

Real-time PCR analysis was performed to analyze the expression pattern of the *HblMYC* gene members with *HbACTIN* (NCBI accession number: JF270598) [21] and *HbRH8* (NCBI accession number: HQ323244) [22] for double housekeeping gene analysis. The primer sequences are listed in Table S1. The real-time PCR reactions and correct analysis were performed with a CFX96/384 touch real-time PCR detection system (Bio-Rad Labratories Inc, Ca, USA). The stability value of the housekeeping genes was assessed. The mean CV was 0.1455 (requirement, <0.25), and the mean M value was 0.4172 (requirement, <0.5). The amplification efficiency of the tested genes was similar to that of the reference genes. Real-time PCR was performed by using 100 ng first-strand cDNA, 10 pmol forward and reverse primers and the TransStart Tip Green® qPCR Super Mix (a SYBR green mix, TransGen Biotech, Beijing, China) in a 20-μL reaction mixture, according to the manufacturer's protocol. Amplification was carried out at 95 °C for 5 min, followed by 40 cycles (95 °C for 10 s, Tm for 20 s, 72 °C for 10 s). The relative expression values were calculated from three biological replicates using a modified $2^{-\Delta\Delta CT}$ method. The single factor variance analysis of the real-time PCR data was performed with SPSS 20 software.

*2.9. Statistic Analysis of Real-Time PCR Analysis Data*

The single factor variance analysis of the real-time PCR data was performed with SPSS 20 software. The histograms were using the value of gene relative expression and drawn by the software of Origin 8.5.1. The heat map was drawn with the software of HemI 1.0 (Heatmap Illustrator) and MeV 4.9 (Multiple Experiment Viewer).

## 3. Results

*3.1. Identification of HblMYC Gene Family*

The *HblMYCs* that were homologous to the MYC family in plants were deduced on the basis of the transcriptome and genome data in rubber clone CATAS 7-33-97 [17,18]. A total of 32 putative *HblMYC* genes, named *HblMYC1-32*, were cloned and validated by RT-PCR and RACE techniques (Primers in Table S1). Firstly, we monitored the physical and chemical characteristics of 32 *HblMYC* genes. The coding sequence lengths of 32 *HblMYC* genes were ranged from 810 to 2613 bp, and the deduced protein lengths were ranged from 269 to 870 amino acids (Table S3). All the predicted *HblMYC* proteins had an HLH-MYC_N domain in the N terminal, an HLH domain in the C terminal. The predicted molecular weights (MWs) of MYC proteins ranged from 30.13 kDa to 94.79 kDa, and the isoelectric points (IPs) changed from 5.10 to 9.26. Most of the *HblMYC* proteins (93.75%) were predicted to be acidic proteins. Analysis of the instability index (II) and GRAVY showed that most of the *HblMYC* proteins were unstable and hydrophilic. In addition, the prediction of subcellular localization showed that all *HblMYC* proteins had at least one nuclear localization signal (NLS) and were located in the nucleus (Figure S1 and Table S3).

*3.2. Phylogenetic Tree and Gene Structures of HblMYCs*

A neighbor-joining tree between 81 MYCs of dicotyledonous (from 17 plant species) and 73 monocotyledonous (from 5 plant species), was structured by the software of MEGA7.0. Our 32 *HblMYCs* were clustered into all seven groups, as follows: only *HblMYC19* within group 1; four *HblMYCs*(*8/17/22/28*) within group 2; *HblMYC24/30* within group 3; *HblMYCs27* within group 4; five *HblMYCs*(*10/13/29/31/32*) within group 5; five *HblMYCs*(*7/9/14/20/25*) within group 6; and fourteen *HblMYCs* within group 7 (Figure 1). The 32 HblMYCs distributed among different groups suggesting that they were diverse in structure and function.

*3.3. Gene Structures and Conserved Motifs of HblMYCs*

The exon–intron distribution of the 32 *HblMYC* genes was analyzed by GSDS tool (Figure 2). The 32 *HblMYCs* were different in the presence and number of introns. There was no intron in 12 *HblMYCs*, 1 intron in *8 HblMYCs* and more than 5 introns in the other *12 HblMYCs* in group 1/2/3 and 5 in the phylogenetic tree. The sequence diversity in the intron of 32 *HblMYCs* suggested that the *HblMYC* gene family may have experienced extensive domain shuffling after genome duplication.

Using MEME software, 10 conserved motifs were identified among the 32 *HblMYCs* (Figure 2 and Table S4). The motif number of the 32 *HblMYC* proteins varied widely, ranging from 3 to 10. Prediction by the Conserved Domains of NCBI annotated motif 1 as the basic Helix Loop Helix (bHLH) domain (cl00081) (Stevens, 2008), motifs 2, 3, 4, 6 and 7 as bHLH-MYC and R2R3-MYB transcription factors N-terminal (cl16716). Motifs 5, 7, 8, 9 and 10 were not annotated in the database. All the 32 *HblMYCs* were contained the basic Helix Loop Helix (bHLH) domain (cl00081) conserved motifs 1 and the bHLH-MYC and R2R3-MYB transcription factors N-terminal conserved motifs 4, indicating that all 32 *HblMYCs* contained the characteristic structure of MYC protein. In addition, motif 8 was unique to the 5 *HblMYCs* (*HblMYC6/11/15/16/21*) in group 7, and which contained all 10 motifs. The motif 3 was unique to almost all HblMYCs (except HblMYC24), and the motif 2 was unique to most of *HblMYCs* (75%, except the *HblMYCs* in group1/2/3). Based on these results, motifs 1, 3 and 4 were relatively conserved in the MYC gene family during evolution in the rubber tree (Figure 2).

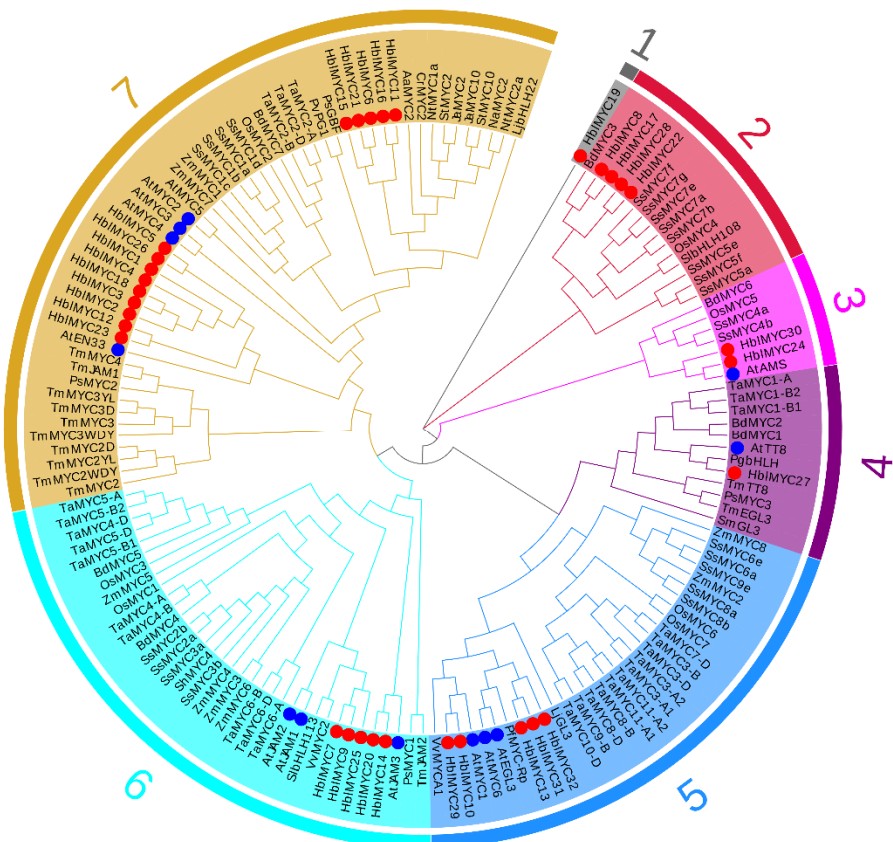

**Figure 1.** Phylogenetic relationship of *HblMYCs*. The MEGA7.0 with the neighbor-joining method was used to conduct the phylogenetic tree. Different numbers and background colors represented different groups of MYC proteins. Red points represented *Hevea brasiliensis*, and blue points represented Arabidopsis. The Latin abbreviations corresponding to species, 5 plant species of monocotyledonous: Bd, *Brachypodium distachyon*; Os, *Oryza sativa*; Ss, *Saccharum spontaneum*; Sh, Saccharum spp. hybrid R570; Ta, *Triticum aestivum*; Zm, *Zea mays*. In addition, 17 plant species of dicotyledonous: Aa, *Artemisia annua*; At, Arabidopsis; Bo, *Brassica oleracea*; Cb, *Chrysanthemum boreale*; Cr, *Catharanthus roseus*; Hb, *Hevea brasiliensis*; Le, *Lithospermum erythrorhizon*; Lj, *Lotus japonicas*; Mt, *Medicago truncatula*; Nt, *Nicotiana tabacum*; Pf, *Perilla frutescens*; Pg, *Punica granatum*; Ps, *Picea sitchensis*; Ps, *Pisum sativum*; Pt, *Populus tomentosa*; Pv, *Phaseolus vulgaris*; Rc, *Ricinus communis*; Sl, *Solanum lycopersicum*; Sm, *Salvia miltiorrhiza*; Sm, *Selaginella moellendorffii*; St, *Solanum tuberosum*; Tc, *Taxus cuspidata*; Vv, *Vitis vinifera*.

*3.4. Promoter Analysis of the HblMYC Genes*

The light, plant growth/development, hormone, and stress-related *cis*-regulatory elements (CREs) were present in the promoter regions of the 32 *HblMYCs* (Figure 3 and Tables S5–S7). The light responsive elements were present in all 32 *HblMYCs*. The jasmonate acid (JA) and abscisic acid (ABA) responsive elements existed in 20 *HblMYCs*. In addition, gibberellin (GA), auxin (IAA), and salicylic acid (SA) responsive elements were found in 10, 7, and 9 *HblMYCs*, respectively. With respect to stress responsive elements, 25 *HblMYCs* had anoxic-induction elements (ARE), 21 *HblMYCs* had wound-induction response elements (WUN-motif). In addition, elements of MBS (MYB binding sites involved in drought-inducibility); DRE (dehydration responsive element); LTR (low-temperature responsive elements); TC-rich repeats (defense and stress response elements); and GC-motifs (enhanced anoxic-specific inducibility) were found in 12, 11, 11, 9, and 4 *HblMYCs*, respectively. The CAT-box motif was a *cis*-acting regulatory element related to meristem expression. It was present in 12 *HblMYCs*. The AACA or GCN4 motif was involved in endosperm-specific expression. It was present in 10 *HblMYCs*. Nine *HblMYCs* contained a circadian motif, which was involved in circadian control. Three *HblMYCs* (*HblMYC1/3/12*) contained a HD-Zip 1 motif, which was involved in endosperm-specific negative expres-

sion. These results indicate that the *HblMYCs* may be widely involved in various biological processes in rubber tree.

**Figure 2.** Phylogenetic tree, gene structure and conserved motif of the *HblMYCs*. Different colors on the phylogenetic tree represented different groups of the *HblMYC* gene family. In exon-intron analysis, the black lines represented introns, the pink boxes represented the coding sequences, and the blue boxes represented the non-coding sequences. In the motif pattern, the motif number 1–10 were displayed in different colored boxes.

### 3.5. Tissue-Specific Expression Patterns of HblMYC Genes

Except for *HblMYC18*, that had little expression in all of the tested eight tissues, the expression of the others could be detected in at least one tissue (Figure 4). First of all, 6 *HblMYCs* (*HblMYC5/17/20/24/26/27*) had a relatively high level while the expression level of 7 *HblMYCs* (*HblMYC1/2/3/4/7/18/25*) was very low in the six tissues (root, stem, leaf, flower, perisperm and bark). Of the six tissues, most of the *HblMYCs* had relatively high expression levels in roots and leaves, but a low level in the perisperm. Secondly, the expression patterns of 32 *HblMYCs* were obviously different between the cambium and the latex. The transcripts of 18 *HblMYCs* (*HblMYC1/2/3/4/5/6/7/9/11/14/15/16/20/21/26/27/30/32*) were relatively abundant in the latex but less in the cambium, while the reverse was true for nine *HblMYCs* (*HblMYC10/13/19/22/23/25/28/29/31*). The abundance of five *HblMYCs* (*HblMYC8/12/17/18/24*) in the two tissues was similar. It was noted that the expression level of four *HblMYCs* (*HblMYC1/2/3/4/6*) in the latex was 20 times that of other tissues. Moreover, the transcripts of four *HblMYCs* (*HblMYC1/2/3/4*) in the latex were more than 14,796, 63,154, 2046 and 450 times that in the cambium, respectively.

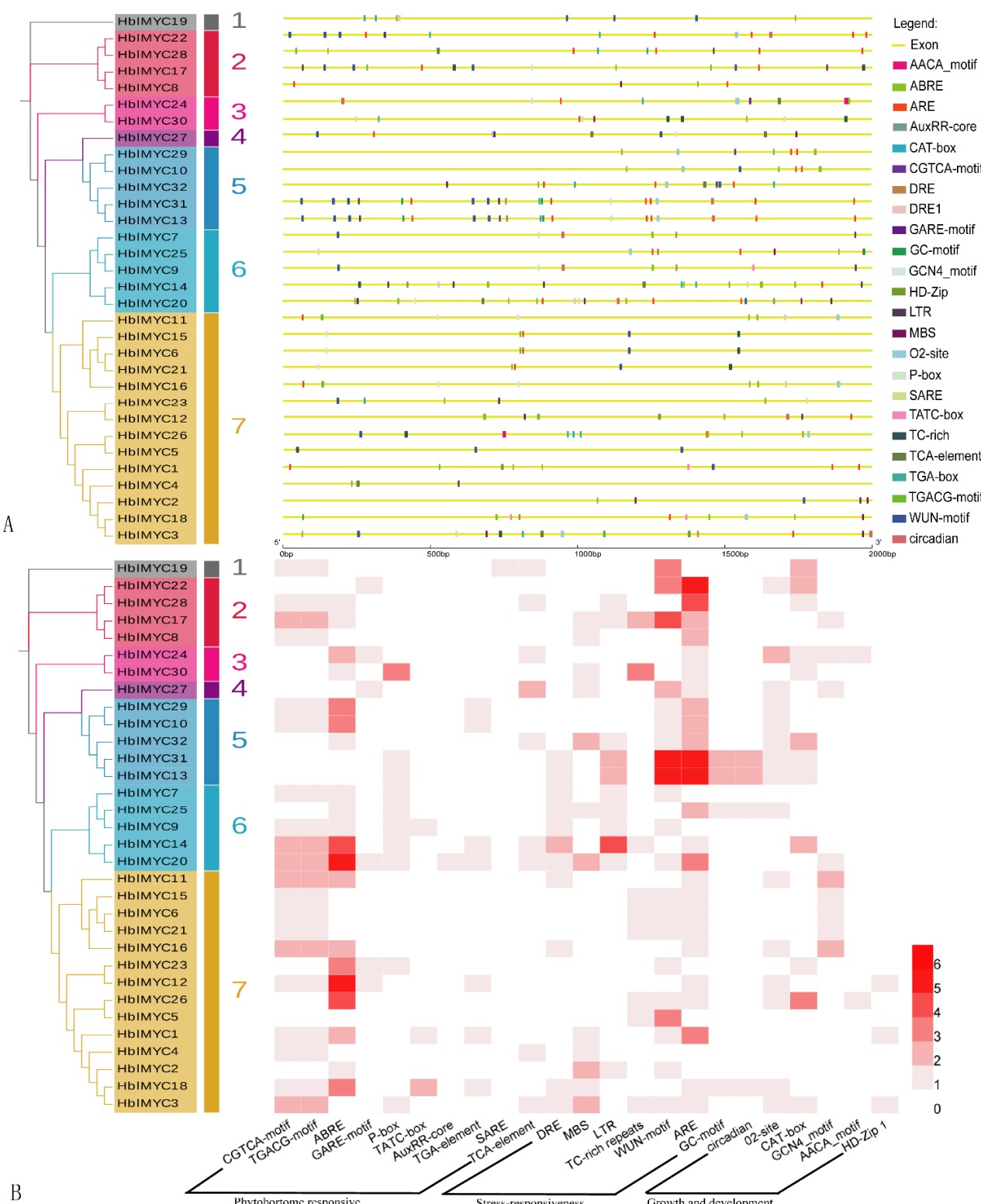

**Figure 3.** *Cis*-regulatory element (CRE) analysis of the *HblMYCs*. (**A**) The location distribution of different CREs in the 2000 bp promoter of 32 *HblMYCs*. Yellow line is the 2000 bp promoter region, different color boxes correspond with the different kinds of CREs; (**B**) The heatmap of CREs in the promoter of 32 *HblMYCs*. The different red boxes show the number of CREs, and white boxes represent that there is no corresponding CRE. Different colors and number on the phylogenetic tree represent different groups of the *HblMYC* gene family.

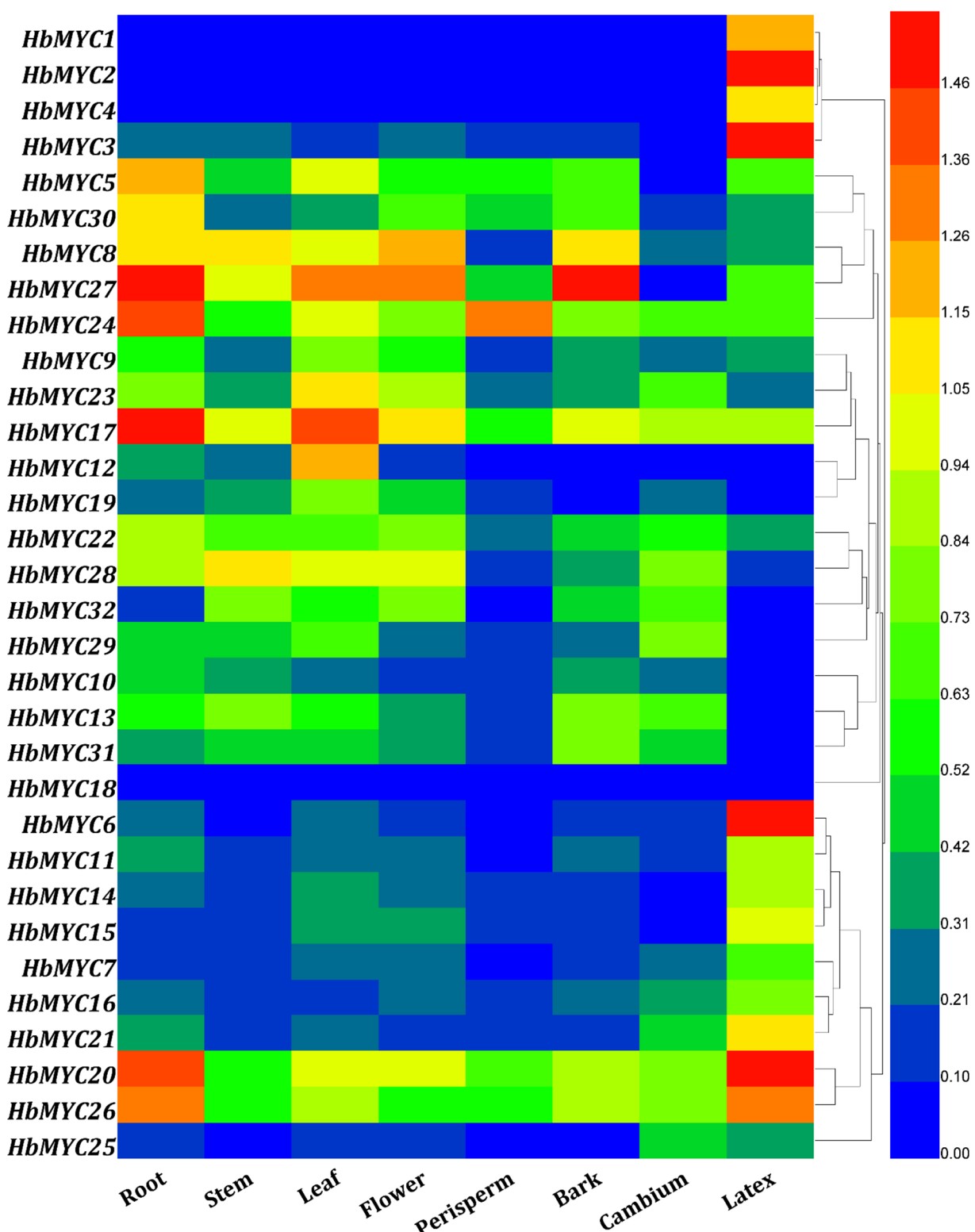

**Figure 4.** Tissue-specific expression of the *HblMYCs*. Eight typical tissues, i.e., feeder root (Root); stem at bronze stage (Stem); mature leaf (Leaf); female flower (Flower); perisperm of seed (Perisperm); trunk bark (Bark); cambium region of stem at stationary phase (Cambium); and latex of virgin tree (Latex) of rubber tree clone CATAS7-33-97 were collected without any treatments at the same time. Each tissue samples from a mixture of three trees. Values are means ± SD of three replicates. The heat maps were drawn by the software of HemI 1.0.

### 3.6. Expression Patterns of HblMYCs in Response to Coronatine (COR)

All the 32 *HblMYC* genes were up-regulated by COR in cambium (Figure 5). The expression patterns could be roughly classified into five groups. Group one included 13 *HblMYCs* (*HblMYC4/5/7/17/18/20/21/23/24/26/29/30*). They were rapidly and shortly up-regulation at 1 h. Group two contained seven *HblMYCs* (*HblMYC3/6/11/15/16/25/31*). They were continuously up-regulated in early time (0.5–4 h). Group three had three *HblMYCs* (*HblMYC19/28/32*). They were only up-regulated at 4 h. Group four had three *HblMYCs* (*HblMYC1/2/27*). They were up-regulated at both early (1–4 h) and late stages (3 d). Group five had six *HblMYCs* (*HblMYC8/9/10/12/13/14*). They were continuously up-regulated in the late stages. The co-expression members may have a certain connection in the same biological processes. In comparison with control, COR caused the increase in the transcript levels of 26 *HblMYCs* more than 10 times than that of the control. It occurred at 1 h for 13 *HblMYCs* (*HblMYC4/5/7/17/18/20/21/23/24/26/29/30*), 2 h for 5 *HblMYCs* (*HblMYC6/9/11/15/25*) and 4 h for 5 *HblMYCs* (*HblMYC19/28/29/31/32*).

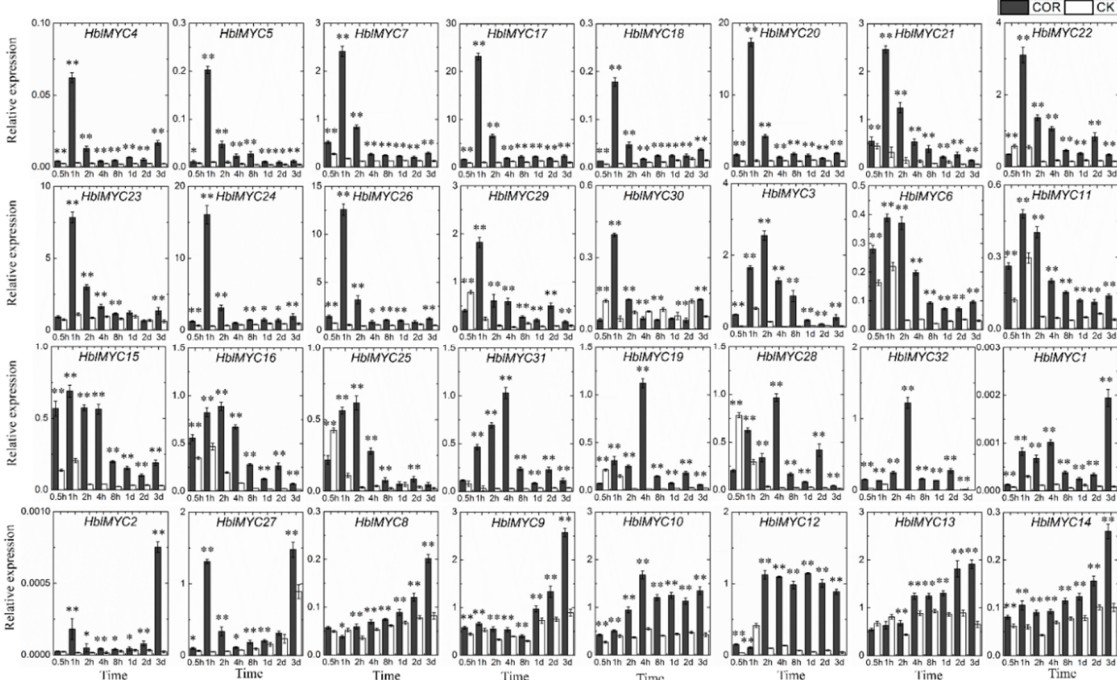

**Figure 5.** Effect of COR on the expression *of HblMYCs* in cambial region of *Hevea* shoots. Values were means ± SD of three replicates. The asterisks (*) and (**) respectively represented significant difference ($p < 0.05$) and very significant difference ($p < 0.01$).

### 3.7. Expression Patterns of HblMYCs in Response to Tapping

Only 5 *HblMYCs* (*HblMYC2/9/18/26/27*) were down-regulated in their gene expression all the time while 27 *HblMYCs* were up-regulated in their gene expression after tapping treatment (Figure 6). The up-regulated expression patterns could be classified into five groups. Group one included seven *HblMYCs* (*HblMYC1/3/4/8/14/15/31*). They were down-regulated at early stages (0.5–6 h), and thereafter up-regulated continuously. Group two contained six *HblMYCs* (*HblMYC5/6/10/11/12/13*). They were down-regulated at the early stage, up-regulated in the middle, and gradually recovered at the late stage. Group three had eight *HblMYCs* (*HblMYC16/17/19/21/22/25/30/32*). They were up-regulated at the early stage, down-regulated in the middle, and again up-regulated at the late stage. Group four had two *HblMYCs* (*HblMYC7/28*) that were up-regulated at all the time intervals. Group five had three *HblMYCs* (*HblMYC20/23/24*). They were up-regulated at the early stage, and thereafter down-regulated. In comparison with the control, tapping caused an increase in the expression level of 19 *HblMYCs* more than two times

that of the control. Thereinto, *8 HblMYCs* (*HblMYC16/17/19/21/22/23/25/32*) at 0.5 h, 3 *HblMYCs* (*HblMYC1/6/10*) at 6 h, and 6 *HblMYCs* (*HblMYC5/6/11/12/13/31*) at 12 h. The *HblMYC19* was up-regulated 9 times at 0.5 h, and *HblMYC12/13* was up-regulated 6 times at 12 h. However, *HblMYC2/9/18* was down-regulated at any time intervals and even lower than 2 times at 0.5 h to 6 h.

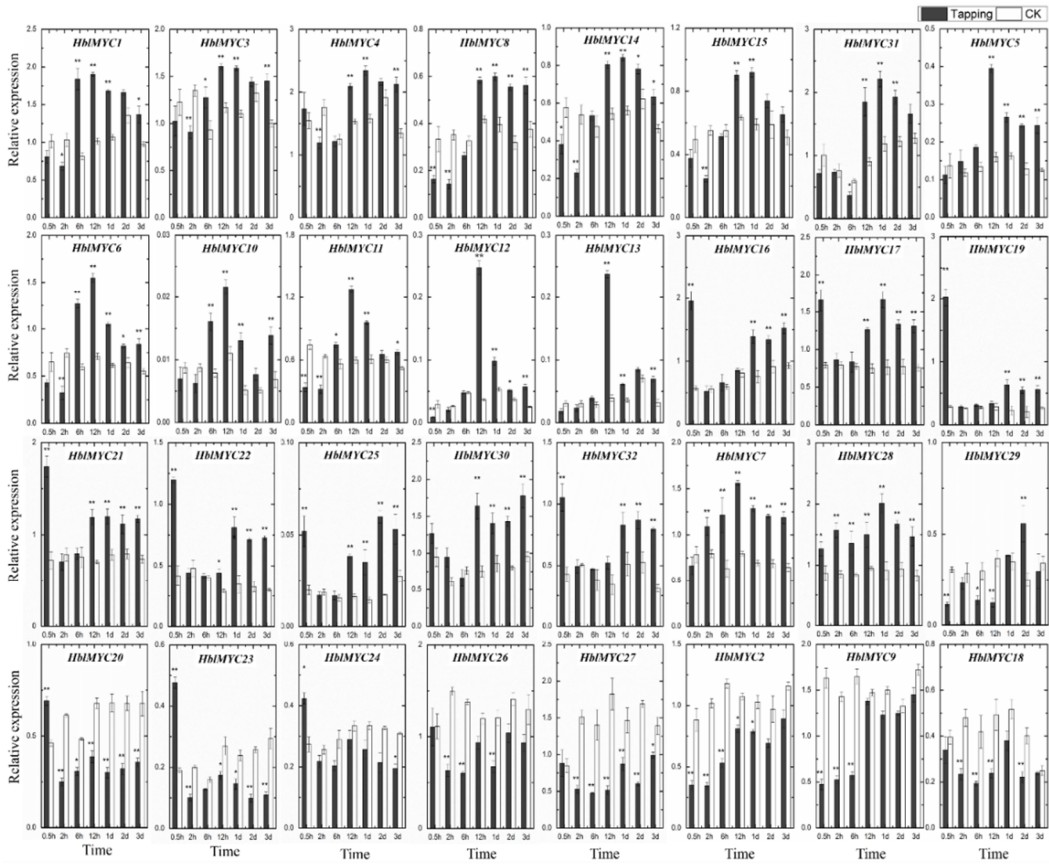

**Figure 6.** Effect of tapping on the expression of *HblMYCs* in latex. Values for real-time PCR analysis were means ± SD of three replicates. The asterisks (*) and (**) respectively represented significant difference (*p* < 0.05) and very significant difference (*p* < 0.01).

## 4. Discussion

The bHLH TFs participate in various biological processes, such as plant fertility [23]; anthocyanin accumulation [24]; leaf senescence [25]; and responses to biotic and abiotic stresses [26]. Although the bHLH gene family has been widely identified in different plant species, such as 147 bHLHs in Arabidopsis [27], 183, 231 and 571 bHLHs in rice, maize and wheat, respectively [28], 94 *VvbHLHs* in grape [29], 95 *GjbHLHs* in *Gardenia jasminoides* [30], and 85 *GbbHLHs* in *Ginkgo biloba* [31], the genome-wide identification of the MYC gene family is limited, including 27 *TaMYCs* in wheat (Chinese Spring) [19]; 26 *TaMYCs* in the wheat genome [32]; and 23 *SsMYCs* in *Saccharum spontaneum* [20]. Here, 32 *HblMYCs* were genome-wide identified in the rubber tree. All MYC members in plant species were characterized by an HLH-MYC_N domain in the N-terminal, a HLH (basic helix-loop-helix) domain in the C-terminal and at least one nuclear localization signal (NLS) [1]. Among the 32 *HblMYCs*, 27 members were novel (Table S1). The *HblMYC1* and *HblMYC2* were the same as the *HblMYC1* and *HblMYC2* [21] and the partial sequences of *HbMYC424* and *HbMYC771* [16]. The *HblMYC16* was the same as *HbMYC2b* [15]. HblMYC18 was the same as the partial sequence of *HbMYC94937*. The *HblMYC21* was the same as HbMYC2 [14]. Large differences in tissue-specific expression, and the diversity in gene structure and the presence of various *cis*-regulatory elements, suggest that the *HblMYCs* participate in regulating various development and stress responses in rubber tree. Most of all, we should

pay more attention to the *HblMYCs* in jasmonate signaling because available data showed that jasmonate signaling plays a pivotal role in regulating laticifer differentiation [13] and rubber biosynthesis [14] in rubber tree.

The transcriptional activation of FPS and SRPP genes by *HblMYC21*, i.e., *HbMYC2* [14] and the small rubber particle protein gene by HblMYC16, i.e., HbMYC2b [15] has been testified. It is noted that *HblMYC6*, *HblMYC11* and *HblMYC15* were closely clustered together with *HblMYC16* and *HblMYC21* (Figure 1) and exhibited similar expression patterns upon tapping, suggesting that the three MYC members may also be involved in regulating rubber biosynthesis. Moreover, the five MYC members were also closely clustered with *AaMYC2* and *CrMYC2*, which positively regulates artemisinin biosynthesis in *Artemisia annua* [33] and alkaloid biosynthesis in *Catharanthus roseus* [34]. On the other hand, the down-regulated MYC members, such as *HblMYC2*, *HblMYC18*, *HblMYC26*, *HblMYC27* and *HblMYC29*, are possibly negative regulators of rubber biosynthesis.

It is well known that jasmonates are the key signal molecules for inducing the differentiation of secretory tissues, such as the resin duct in the xylem of Pinaceae plants [35] and the laticifer in rubber trees [6,7,9–11,13]. Given the essential component of MYCs in JA signaling, some of the *HblMYCs* should participate in the regulation of laticifer differentiation. However, the involvement of *HblMYCs* in regulating laticifer differentiation has not ever been reported and all of the 32 *HblMYCs* are up-regulated in the cambium region by COR, the jasmonate mimic, although the expression patterns are different to some extent. Tentatively, *HblMYC24* and *HblMYC30* may be related to laticifer differentiation upon COR. Firstly, they were rapidly up-regulated at an early stage of response to COR; secondly, they were clustered together with *AtAMS*, which is a development-related MYC member and plays a crucial role in pollen development [36]; thirdly, the CAT-box that is a present CRE in the BjuA07.CLV1, and necessary for shoot and floral meristem development [37], was also present in the promotor of *HblMYC24* and *HblMYC30*.

## 5. Conclusions

The 32 MYC genes were identified genome-wide in the rubber tree. The diversity in structure and the presence of various *cis*-regulatory elements implies the involvement of *HblMYCs* in various development and stress responses. Based on the expression patterns and phylogenetic relationship with the MYCs that have been functionally identified in other plants, the *HblMYC24* and *HblMYC30* may be related to laticifer differentiation while the *HblMYC6*, *HblMYC11* and *HblMYC15*, as well as *HblMYC16* and *HblMYC21*, may positively regulate rubber biosynthesis. The results provide a foundation for understanding molecular mechanism of jasmonate signaling in regulating laticifer differentiation and rubber biosynthesis in rubber tree.

**Supplementary Materials:** The following supporting information can be downloaded at: https://www.mdpi.com/article/10.3390/f13040531/s1, Figure S1. Multiple sequence alignment of 32 HblMYC proteins; Table S1. The primers of 32 *HblMYC* genes in this study; Table S2. Amino acid sequence of 32 HblMYC proteins and other 122 MYC proteins. Table S3. Identification and characterization of the 32 *HblMYC* genes; Table S4. Ten predicted conserved motifs of the 32 HblMYC proteins; Table S5. The location information of cis-regulatory elements (CREs) in the *HblMYC* promoter region; Table S6. The number of cis-regulatory elements in the *HblMYC* promoter region; Table S7. Promoter elements and its function annotation of the 32 *HblMYC* genes.

**Author Contributions:** Conceived and designed the experiments: W.-M.T. and S.-X.Z.; Performed the experiments: S.-X.Z., S.-H.W., S.-G.Y. and J.B.; Analyzed the data: S.-X.Z., J.-Q.C. and J.B.; Contributed reagents/materials/analysis tools: S.-X.Z.; Wrote the manuscript: S.-X.Z. and W.-M.T. All authors have read and agreed to the published version of the manuscript.

**Funding:** This work was supported by the Central Public-interest Scientific Institution Basal Research Fund for Chinese Academy of Tropical Agricultural Sciences (No.1630022021001), the National Natural Science Foundation of China (31800577), the National Key R&D Program of China (Grant

No. 2018YFD1000502), the Fund for Modern Agro-industry Technology Research System (Grant No. CARS-33-YZ1), the Natural Science Foundation of Hainan Province (317278).

**Conflicts of Interest:** The authors declare that they have no known competing financial interests or personal relationships that could have appeared to influence the work reported in this paper.

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
