# Peer review of "Genome-Wide Identification and Expression Analysis of MYC Transcription Factor Family Genes in Rubber Tree (Hevea brasiliensis Muell. Arg.)"

_forests, doi:10.3390/f13040531_

Round 1

Reviewer 1 Report

  1. The manuscript under review, “Genome-wide identification and expression analysis of MYC 2 transcription factor family genes in rubber tree (Hevea brasili- 3ensis Muell. Arg.)s” was performed by the authors found to be appreciable.

    The manuscript presents the results of a series of experiments investigating the identification and expression analysis of MYC 2 transcription factor family genes in rubber trees.

    They used two treatments (COR and Tapping) to increase the expression of transcripts. The COR was used at 20 ug/L and the other treatment was tapping.

    The topic is timely adds to our understanding of the identification of MYC transcription factors and their importance in regulating the secondary laticifer differentiation and rubber biosynthesis in rubber trees.

    Although I have specific comments, the work was carefully executed, which prevented me from recommending this paper for publication in its current form. However, with a significant amount of effort, the authors should be able to address most of these concerns, and once that is done, the paper may be acceptable for publication.

    Specific comments are given below, and authors should carry out constructive suggestions to improve the quality of this manuscript.

    1. The introduction could describe why they use those treatments or mention previous work-related.
    2. Figure 4 should be appropriately described with more details for the tissue.
    3. The discussion did not mention a relation between the expression pattern and plant tissue. Also, the authors did not remark on the difference between treatments.

Author Response

Thank you for the reviewers comments concerning our manuscript, entitled“Genome-wide identification and expression analysis of MYC transcription factor family genes in rubber tree (Hevea brasiliensis Muell. Arg.)” (ID: forests-1618596). Those comments are all valuable and very helpful for revising and improving our paper, as well as the important guiding significance to our researches. We have revised the content against the three opinions, added summary of previous work in introduction, and described more details for the tissues in figure 4, and added information on tissue-specific expression correlating with MYC gene function diversity in discussion. Please see the attachment.

Reviewer 2 Report

MYC transcription factors play important roles in jasmonic acid signaling networks. This work by Zhang et al. conducted gene expression analysis of MYC transcription factors in rubber tree. Although authors performed in silico analysis of rubber tree MYCs plus gene expression analysis, there is no fundamental questions tackled in this manuscript. The assays of molecular and functional characterization of the MYCs identified in rubber trees were not conducted in this work. These problems bring the difficulties in justifying publication of this work in Forests. 

Author Response

Thank you for the reviewers comments concerning our manuscript, entitled“Genome-wide identification and expression analysis of MYC transcription factor family genes in rubber tree (Hevea brasiliensis Muell. Arg.)” (ID: forests-1618596). Those comments are all valuable and very helpful for revising and improving our paper, as well as the important guiding significance to our researches. At present, the transgenic and gene editing in rubber tree is still difficult to research, a few papers about which have been published, but the transgenic technology platform of rubber tree is still not establish. Although a lot of works were performed, low transformation efficiency is current status and cannot meet the need of gene function homology verification. The model plants, such as Arabidopsis, tobacco and poplar, are lack of laticifer tissue, and cannot be used to detect the gene function of secondary laticifer differentiation. In the future, we will transform the candidate HblMYC genes screened in this paper into Taraxacum kok-saghyz for gene function verification. Please see the attachment.

Reviewer 3 Report

This is an important study submitted by one of the leading experts on rubber tree genome analysis. The manuscript focused on the identification and characterization of 32 genes belonging to the MYC transcription factor family in rubber tree, the primary natural source of rubber production. Expression analysis of genes in response to coronatine or tapping in cambium or latex, respectively, allows the identification of potential genes related to laticifer differentiation and rubber biosynthesis.

Some recommendations:

Line 16: COR; Please indicate the meaning of the abbreviation

Line 29: JA; * Please indicate the meaning of abbreviation

Line 63; * Please include the abbreviation of extension unit

Line 89: Please remove the partial sequence:” genomic DNA was isolated from

Legend of Figure 1: Please indicated scientific names in italics

Line 206: There, should be The?

Line 321: Please delete is

Line 254: Please inserts a space between “had3”; had 3

Line 291: Please write Grape un lowercase letter (grape)

Line 308: Please insert a space

 Lines 43 and 44: to facilitate the comprehension to the readers, please explain abbreviations of FPS and SRPP

Author Response

Thank you for the reviewers comments concerning our manuscript, entitled“Genome-wide identification and expression analysis of MYC transcription factor family genes in rubber tree (Hevea brasiliensis Muell. Arg.)” (ID: forests-1618596). Those comments are all valuable and very helpful for revising and improving our paper, as well as the important guiding significance to our researches. We have revised the manuscript word by word, and provided point-by-point responses to the requirements.

  1. The full name of COR was added in line 16.
  2. The full name of JA was added in line 29.
  3. The full name of EU was added in line 63.
  4. ”genomic DNA was isolated from” was removed in line 89.
  5. The italic of scientific names was revised in legend of Figure 1.
  6. “There” was revised in line 206.
  7. “is” was revised in line 321.
  8. A space between “had3” was added in line 254.
  9. “Grape” was revised in line 291.
  10. A space was added in line 308.
  11. The full name of FPS and SRPP were added in line 43 and 44.
